



# Retrieval of Atmospheric CFC-11 and CFC-12 from High-resolution FTIR Observations at Hefei and Comparisons with Satellite Data

Xiangyu Zeng[1,2], Wei Wang[1], Cheng Liu[1,3,4,5,6], Changgong Shan[1], Yu Xie[7], Peng Wu[1,2], Qianqian Zhu[1,2], Alexander Polyakov[8]

[1]Key Laboratory of Environmental Optics and Technology, Anhui Institute of Optics and Fine Mechanics, Hefei Institutes of Physical Science, Chinese Academy of Sciences, Hefei, 230031, China

[2]University of Science and Technology of China, Hefei, 230026, China

[3]Department of Precision Machinery and Precision Instrumentation, University of Science and Technology of China, Hefei, 230026, China

[4]Center for Excellence in Regional Atmospheric Environment, Institute of Urban Environment, Chinese Academy of Sciences, Xiamen, 361021, China

[5]Key Laboratory of Precision Scientific Instrumentation of Anhui Higher Education Institutes, University of Science and Technology of China, Hefei, 230026, China

[6]Anhui Province Key Laboratory of Polar Environment and Global Change, University of Science and Technology of China, Hefei, 230026, China

[7]Department of Automation, Hefei University, Hefei, 230601, China

[8]Faculty of Physics, Saint Petersburg State University, Saint Petersburg, 199034, Russia

*Correspondence to*: Wei Wang (wwang@aiofm.ac.cn), Cheng Liu (chliu81@ustc.edu.cn)

**Abstract.** Synthetic halogenated organic chlorofluorocarbons (CFCs) play an important role in stratospheric ozone depletion, and contribute significantly to the greenhouse effect. In this work, the mid-infrared solar spectra measured by ground-based high-resolution Fourier transform infrared spectroscopy (FTIR) were used to retrieve atmospheric CFC-11 ($CCl_3F$) and CFC-12 ($CCl_2F_2$) at Hefei, China. We implemented a new retrieval strategy, and analyzed the retrieval errors. The CFC-11 columns observed from January 2017 to December 2020 and CFC-12 columns from September 2015 to December 2020 show a similar annual decreasing trend and seasonal cycle, with an annual rate of $(-0.47\% \pm 0.16)$ % yr$^{-1}$ and $(-0.79 \pm 0.31)$ % yr$^{-1}$, respectively. CFC-11 total columns were higher in summer, and CFC-12 total columns were higher in summer and autumn. Both of CFC-11 and CFC-12 total columns reached the lowest in spring. The annual decreasing rate of near-surface concentration is $(-0.60 \pm 0.26)$ % y$^{-1}$ for CFC-11, and $(-0.81 \pm 0.25)$ % y$^{-1}$ for CFC-12. So the decline rate of CFC-11 is significantly lower than that of CFC-12. Further, FTIR data were compared with the ACE-FTS satellite data, WACCM (Whole Atmosphere Community Climate Model) data and the data from other NDACC (Network for the Detection of Atmospheric Composition Change) station. The mean relative difference between the vertical profiles observed by FTIR and ACE-FTS is $(-5.6 \pm 3.3)$ % and $(4.8 \pm 0.9)$ % for CFC-11 and CFC-12 for altitude from 5.5 to 17.5 km, respectively. The results demonstrate our FTIR data agree relatively well with the ACE-FTS satellite data. The annual decreasing rate of CFC-



11 measured from ACE-FTS and calculated by WACCM are (−1.15 ± 0.22) % and (−1.68 ± 0.18) %, respectively. The
interannual decreasing rates of atmospheric CFC-11 obtained from ACE-FTS and WACCM data are higher than that from
FTIR observations. Also, the annual decreasing rate of CFC-12 from ACE-FTS and WACCM is (–0.85 ± 0.15) % and (–0.81
± 0.05) %, respectively, close to the corresponding values from the FTIR measurements. Further, the total columns of CFC-11
observed at the Hefei site are very close to those at St. Petersburg station, with a mean difference of $3.63 \times 10^{12}$ molec·cm$^{-2}$,
while the total columns of CFC-12 are $1.69 \times 10^{14}$ molec·cm$^{-2}$, slightly higher than those at St. Petersburg station.

## 1 Introduction

Synthetic halogenated organic chlorofluorocarbons (CFCs) have been widely used in industry as refrigerants, foam-blowing
agents and propellants, due to their stable and non-toxic chemical properties (Mcculloch et al., 2003). The photolysis of CFCs
in the stratosphere significantly cause the depletion of stratospheric ozone, so CFC-11 ($CCl_3F$) and CFC-12 ($CCl_2F_2$) are
currently classified as important ozone depleting substances (ODSs) (Molina and Rowland, 1974). With the long atmospheric
lifetime, about 52 years for CFC-11 and 102 years for CFC-12, they can be transported to the polar region and accumulated to
cause the polar ozone depletion . CFCs also have high global warming potentials (GWPs), being considered as the greenhouse
gases (Molina et al., 2009). GWP refers to the ratio of radiative forcing for a given mass of a substance relative to $CO_2$ emissions
of the same mass over a given time (Fang et al., 2018). The GWPs of CFC-11 and CFC-12 are reported to be 5160 and 10300
for the 100-year time .
The Montreal Protocol on Substances that Deplete the Ozone Layer came into effect in 1989, for limitation of ozone
depleting substances in industrial products, and to avoid their continued damage to the Earth's ozone layer. China, as one of
the countries with the highest CFCs emissions, has committed to phasing out CFCs productions by 2010 (Wan et al., 2009;
Wu et al., 2018). The atmospheric concentrations of CFCs declined slowly, and the ozone layer began to recover gradually
under the implementation of the ban. However, there was a slowdown in the global declining CFC-11 concentrations after
2012 from the observations at remote measurement sites, and the difference between expectations of accelerated rates of
decline and observations widened from 2012 to 2017, suggesting unreported new productions of CFC-11 (Montzka et al.,
2018). The atmospheric observations at Gosan, South Korea, and Hateruma, Japan, combined with the simulations of
atmospheric chemical transport models showed, there was increase in CFC-11 emissions around Shandong and Hebei
provinces in China from 2014 to 2017 (Rigby et al., 2019). Also, a study based on a Bayesian Parameter Estimation (BPE)
model estimates global unexpected CFC-11and CFC-12 emissions reached 23.2 and 18.3 kg/year during 2014-2016 (Lickley
et al., 2021). Meanwhile, The atmospheric measurements and simulations at Mauna Loa Observatory, Gosan, South Korea and
Hateruma, Japan show that, CFC-11 emissions in China decreased after 2019, and the decline of the global average CFC-11
concentrations accelerated (Montzka et al., 2021; Park et al., 2021).
Study of the temporal-spatial distribution and variations of CFCs in the atmosphere is of great significance to reduce
stratospheric ozone depletion and greenhouse gas emissions. In recent decades, in-situ and remote sensing techniques have



been used to monitor CFCs (Khosrawi et al., 2004; Eckert et al., 2016; Kellmann et al., 2012; Lin et al., 2019; Zhang et al., 2011). The surface in-situ measurements monitor long-term trend and seasonal variations of the target gases, such as those in the Advanced Global Atmospheric Gases Experiment (AGAGE), the World Data Centre for Greenhouse Gases (WDCGG), NOAA's Halocarbons& other Atmospheric Trace Species Group (HATS) (Rigby et al., 2013). In the last decade, the in-situ

CFCs measurements were also performed in many Chinese cities and suburbs (Zhang et al., 2017; Lin et al., 2019; Zhen et al., 2020; Yang et al., 2021; Yi et al., 2021; Benish et al., 2021). In-situ observations provide highly precise atmospheric concentration data. Yi et al. (2021) measured the annual mean mixing ratios of major halocarbons in five different cities in China from 2009 to 2019, and the CFC-11 and CFC-12 concentrations in the atmosphere showed a downward trend (Yi et al., 2021). The in-situ measurements mostly offer the near-surface ambient mixing ratios, and only few measurements are

conducted in the troposphere and stratosphere. Benish et al. (2021) collected air samples in 500 ~ 3500 m by aircraft above Hebei Province in 2016 and found atmospheric CFC-11 and CFC-12 were higher than global tropospheric background levels, and deduced that CFC-11 and CFC-12 has new production in eastern China (Benish et al., 2021).

Satellite remote sensing techniques, such as high resolution dynamics Limb Sounder (HIRDLS), Improved Limb Atmospheric Spectrometer (ILAS), the collocated Michelson Interferometer for Passive Atmospheric Sounding (MIPAS) and

Atmospheric Chemistry Experiment Fourier transform spectrometer (ACE-FTS), are also mainly used to measure the global distribution of CFCs (Eckert et al., 2016; Hoffmann et al., 2014; Kellmann et al., 2012; Khosrawi et al., 2004; Oshchepkov et al., 2006; Tegtmeier et al., 2016; Steffen et al., 2019). In addition, in the study of Chen et al. (2020), global CFC-11 surface concentration and trend are observed by Atmospheric Infrared Sounder (AIRS) aboard the NASA Aqua satellite(Chen et al., 2020). Garkusha et al. (2017) reported modern satellite Fourier spectrometer IRFS-2 instrument has capability to retrieve the

CFC-11 and CFC-12 in the information gathering mode (Garkusha et al., 2017). Airborne remote sensing instruments are also used to measure atmospheric CFCs, such as limb-imaging infrared FTS (Fourier transform spectrometer) GLORIA (Gimballed Limb Observer for Radiance Imaging of the Atmosphere), and limb-scanning infrared FTS MIPAS-STR (Michelson Interferometer for Passive Atmospheric Sounding –STRatospheric aircraft) (Woiwode et al., 2012; Johansson et al., 2018). However, due to the low sensitivity and large measurement error near surface, satellite and airborne remote sensing data need

to be verified by ground-based observations (Mahieu et al., 2005; Eckert et al., 2016).

The ground-based remote sensing Fourier transform infrared (FTIR) spectroscopy is used to detect the vertical profile and long-term trend of trace gases with high precision (Godin-Beekmann, 2007; De Maziere et al., 2018). Notholt (1994) measured atmospheric CFCs at the polar night by ground-based FTIR with the moon as the light source in the 1990s (Notholt, 1994). Mahieu et al. (2010) measured the CFC-11, CFC-12 and HCFC-22 total columns and annual trends above Jungfraujoch station,

Switzerland by FTIR technique (Mahieu et al., 2010). Zhou et al. (2016) observed the vertical profiles and the annual variations of CFC-11, CFC-12 and HCFC-22 at St Denis and Maïdo FTIR sites from 2004 to 2016, and compared with MIPAS/ENVISAT satellite data (Zhou et al., 2016). Prignon et al. (2019) utilized the Tikhonov regularization strategy to improve the retrieval of atmosphere HCFC-22 vertical profiles, observed by FTIR from 1988 to 2017 above Jungfraujoch (Prignon et al., 2019). Polyakov et al. (2021) refined the infrared solar radiation retrieval strategy to estimate the column-averaged dry air mole




fractions of CFC-11, CFC-12 and HCFC-22 at St. Petersburg (Polyakov et al., 2021).

The objective of this paper is to obtain the CFC-11 and CFC-12 vertical profiles and total columns from the solar spectra based on ground-based FTIR spectroscopy. Section 2 describes the Hefei FTIR observing site, the retrieval parameters and retrieval strategy. Then we present the retrieval results and discuss the inter-annual variability and seasonality of CFC-11 and CFC-12, and compare the data with the ACE-FTS satellite data, the WACCM data, and the data from other NDACC station in Section 3. A summary is drawn in Section 4.

## 2 Measurement methods of Atmospheric CFC-11 and CFC-12

### 2.1 Observing site and instruments

The Hefei ground-based FTIR remote sensing site (31.91°N, 117.17°E and 29 m above sea level) is located at the Anhui Institute of Optics and Fine Mechanics, Chinese Academy of Sciences, in the north-western rural area of Hefei city in eastern China, adjacent to a lake in a flat terrain. The instruments include a high-resolution Fourier transform infrared Bruker IFS 125HR spectrometer and a solar tracker (A547N) installed on the roof. A meteorological station (Zeno, coastal environmental systems, USA) on the roof records surface pressure, temperature, relative humidity, wind speed and other meteorological information since September 2015 (Yin et al., 2020; Zhang et al., 2020; Shan et al., 2021b; Shan et al., 2021a; Wang et al., 2017; Yin et al., 2019). The spectrometer uses a liquid-nitrogen-cooled MCT detector combined with a KBr beamsplitter to record the mid-infrared spectra. The mid-infrared solar absorption spectra covering about 800–1200 cm$^{-1}$ are used to retrieve the target gases in this study, with a spectral resolution of 0.005 cm$^{-1}$ and an optical path difference (OPD) of 180 cm.

### 2.2 Retrieval parameter setting

Table 1 lists the parameters used for CFC-11 and CFC-12 retrievals. The retrieval window of CFC-11 are 830-860 cm$^{-1}$, and the spectral window centered at 1161 cm$^{-1}$ were chosen to retrieve atmospheric CFC-12 (Zhou et al., 2016; Polyakov et al., 2021). Atmospheric parameters, such as H$_2$O, temperature and pressure profiles are adopted from National Centers for Environment Protection (NCEP) reanalysis data (Kalnay et al., 1996). The priori profiles of CFCs and interfering gases except H$_2$O are derived from the Whole Atmosphere Community Climate Model (WACCM) version 6, and the priori profiles of CFC-11 and CFC-12 are from the monthly mean of 2017–2020 and 2015-2020 WACCM v6 data, respectively. The spectroscopic line parameters for CFC-11, CFC-12 and COCl$_2$ are calculated based on empirical pseudo-line-lists (PLL), and the line parameters of other interfering gases are provided by HITRAN 2012 (Rothman et al., 2013). According to the study of Polyakov et al. (2021), because the CFC-11 retrieval window is wide, it is necessary to consider the influence of the increase in the thickness of amorphous water ice in the instrument caused by water vapor in the atmosphere (Polyakov et al., 2021). Therefore, the curvature is considered to be used in the retrieval and the uncertainty is set to 10$^{-6}$.





**Table 1. Retrieval parameters used for CFC-11 and CFC-12**

| Species | CFC-11 | CFC-12 |
|---|---|---|
| Spectral range ($cm^{-1}$) | 830–860 | 1160.2–1161.4 |
| Interfering species | $H_2O$, $COCl_2$, $HNO_3$, $CO_2$, $O_3$ | $H_2O$, $O_3$, $N_2O$, $CH_4$ |
| T, P and $H_2O$ profiles | NCEP | NCEP |
| A priori profile | WACCAM v6 | WACCAM v6 |
| Spectroscopy | PLL, HITRAN 2012 | PLL, HITRAN 2012 |
| Background | slope, curvature, zshift, beam | slope |

**2.3 Retrieval strategy**

The total columns and vertical profiles of CFC-11 and CFC-12 are retrieved by the SFIT4 (version 0.9.4.4) algorithm, which implements the optimal estimation method (OEM) (Rodgers and Connor, 2003). The vector of measurement $y$ is described by the forward model $F$ and the state vector $x$ as:

$$y = F(x, b) + \varepsilon \tag{1}$$

the forward model $F(x, b)$ relates the true state of the atmosphere and the observation system, where $\varepsilon$ represents the random noise of measurement and the uncertainty of retrieval, state vector $x$ is unknown, containing vertical profiles of gas and instrument-related parameters to be retrieved, $b$ is a vector including the temperature and pressure profiles, instrument specifications and other information that have impact on measurement vector but not to be retrieved. The retrieved state vector

can be found by the known result $y$. The forward model is nonlinear for FTIR measurement, so the algorithm uses the method of Newtonian iteration to calculate the result of $i$ time:

$$x_{i+1} = x_i + (K_i^T S_\varepsilon^{-1} K_i + S_a^{-1})^{-1} \times \{K_i^T S_\varepsilon^{-1}[y - F(x_i)] - S_a^{-1}(x_i - x_a)\} \tag{2}$$

where $x_a$ is the a priori profile, $K$ is the Jacobian matrix, $S_a$ and $S_\varepsilon$ are the priori covariance matrix and the measurement covariance matrix. The solution of the inverse problem is an ill-posed process constrained by a priori state vector $x_a$ and

regularization matrix $R(R = S_a^{-1})$. In the preliminary study, we applied the OEM regularization to retrieve CFC-12, which is regularized by a diagonal a priori covariance matrix. However, there were obvious oscillations in some retrieved profiles as shown in Fig. 1(b) and Fig. 2(b), resulting in the unreasonable distribution below the stratosphere. According to the study in Prignon et al. (2019) and Vigouroux et al. (2009), OEM regularization may lead to an unrealistic distributions in the retrieved vertical profiles, while Tikhonov regularization constrains the difference between $x - x_a$ to a constant profile to avoid

spurious oscillations (Prignon et al., 2019; Vigouroux et al., 2009; Sussmann et al., 2011).

Tikhonov $L_1$ regularization can be defined to constrain matrix $R = \alpha L_1^T L_1 \in R^{n \times n}$, $\alpha$ is the regularization strength and $L_1$ is the discrete first derivative operator (Tikhonov, 1963). For the constrained matrix transformation $T$ in non-altitude constant retrieval grid, as $R' = \alpha L_1^T T L_1$, $T$ is:





$$\mathbf{T} = \begin{pmatrix} \frac{1}{\Delta z_1^2} & 0 & \cdots & 0 \\ 0 & \frac{1}{\Delta z_2^2} & \ddots & \vdots \\ \vdots & \ddots & & 0 \\ 0 & \cdots & 0 & \frac{1}{z_{n-1}^2} \end{pmatrix} \in \mathbf{R}^{(n-1)\times(n-1)} \tag{3}$$

where $\Delta z$ is the thickness of each layer with index $n$.

The regularization strength $\alpha$ is crucial to constrain the retrieved profiles and extract more information from measurements, so we follow the approach described in Steck (2002) that minimizes the measurement error (Steck, 2002). Using all spectra collected in 2020 to test the regularization strength, the test results are listed in Table 2. CFC-11 has the minimum measurement error of 0.50% for regularization strength $\alpha= 10^2$, while CFC-12 has the minimum measurement error of 0.14% for $\alpha= 10^4$,

and the degrees of freedom for signal (DOFs) of the two gases are greater than 1. So the regularization strength is chosen as $10^2$ and $10^4$ for CFC-11 and CFC-12 retrieval, respectively.

**Table 2. The measurement errors and retrieved DOFs for (a) CFC-11 (b) CFC-12 by using different regularization strength $\alpha$ value.**

| (a) | | | | |
|---|---|---|---|---|
| $\alpha$ | 10 | $10^2$ | $10^3$ | $10^4$ |
| Measurement error (%) | 0.51 | 0.50 | 0.50 | 0.50 |
| DOFs | 1.10 | 1.01 | 1.00 | 1.00 |


| (b) | | | | |
|---|---|---|---|---|
| $\alpha$ | 10 | $10^2$ | $10^3$ | $10^4$ |
| Measurement error (%) | 0.25 | 0.20 | 0.15 | 0.14 |
| DOFs | 2.07 | 1.70 | 1.20 | 1.03 |

**2.4 Spectral retrieval of CFC-11 and CFC-12**

A typical spectrum was analyzed to retrieve CFC-11 and CFC-12, and the spectrum was collected at 01:55:48 UTC on 15 January 2017, with a solar zenith angle of 63.03°. The spectral retrieval window, the retrieved vertical profile, and the total

column averaging kernels for CFC-11 and for CFC-12 are plotted in Fig. 1 and Fig. 2, respectively. The fitting residuals of CFC-11 are within ±2%, and the root-mean-square (RMS) error is 0.309%. The fitting residuals of CFC-12 are within ±1%, and the RMS error is 0.298%. The troposphere vertical distributions of CFC-11 and CFC-12 have obvious oscillations with OEM regularization. CFC-11 reaches the maximum at the height of 10–15 km, CFC-12 has the maximum concentration at 1.5–7.5 km, and then decreases sharply. While the profile of mixing ratio retrieved by Tikhonov regularization method has a

relatively small variation in the troposphere, and CFC-11 and CFC-12 are mainly distributed within 0–20 km. The priori profile



of CFC-12 is similar to the retrieved profile with the Tikhonov regularization, and tropospheric concentrations of the retrieved CFC-11 profile are significantly higher than those of the a priori profile. The total column averaging kernels in Fig. 1(c) and 2(c), describe the sensitivity of the height dependence of the retrieved profile to concentration perturbations at various atmospheric levels. The high sensitivity means the profile retrieved mainly comes from the measured spectrum rather than a priori information (Rodgers and Connor, 2003). The high sensitivity is at 0–40 km for CFC-11 measurements, while the total column averaging kernel is close to 0 above 40 km, which means that the sensitivity is very low. For CFC-12, the total column averaging kernel above 60 km tends to be zero.

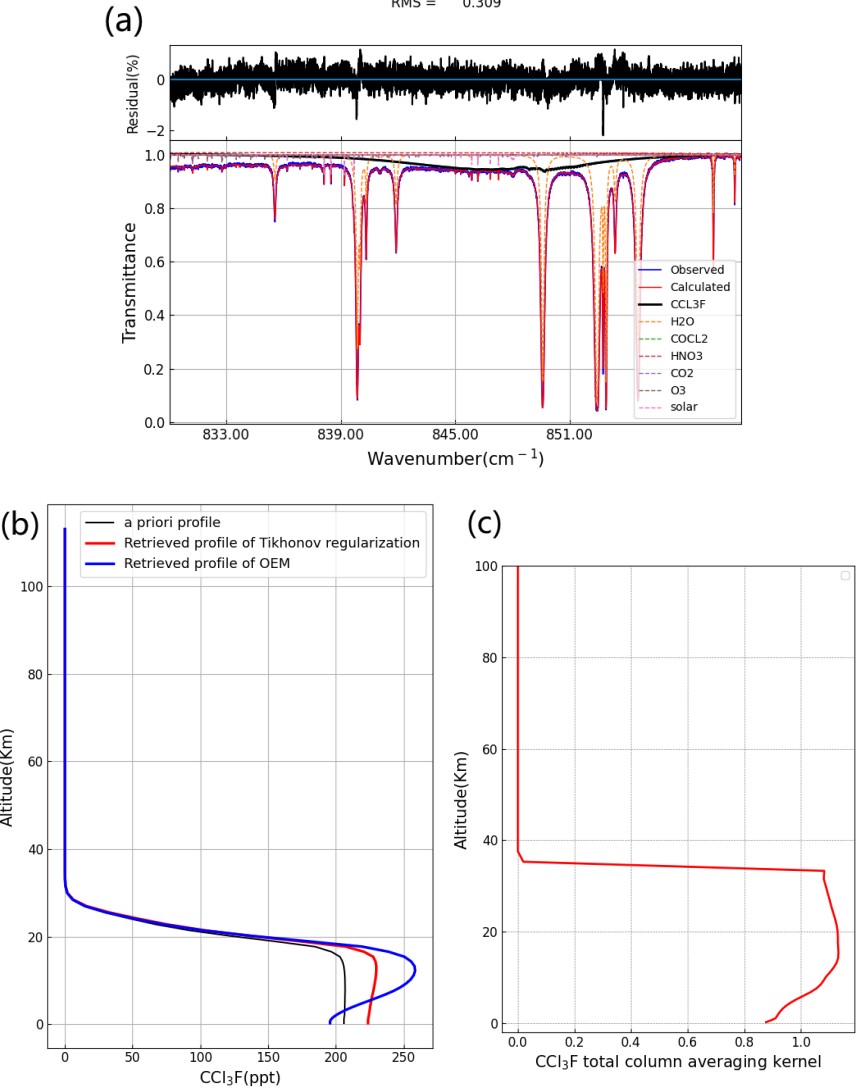

**Figure 1: (a) Measured (blue) and fitted (red) CFC-11 spectrum (01:55:48 UTC 15 January 2017, solar zenith angle of 63.03°) in the 1 microwindow; (b) the CFC-11 profiles, the black line represents a priori profile, the red line represents a retrieved profile using Tikhonov regularization, the blue represents the retrieved profile using OEM; (c) the total column averaging kernel of CFC-11.**





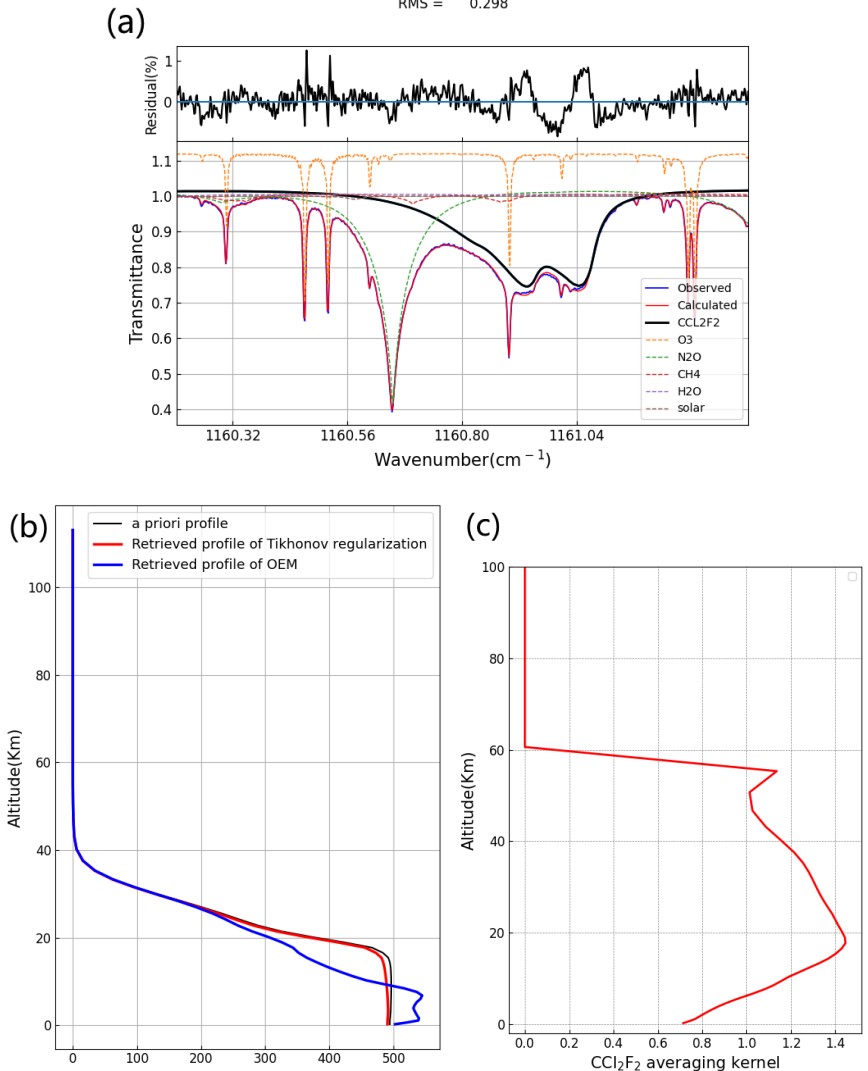

Figure 2: (a) Measured (blue) and fitted (red) CFC-12 spectrum (01:55:48 UTC 15 January 2017, solar zenith angle of 63.03°) in the 1 microwindow; (b) the CFC-12 profiles, the black line represents a priori profile, the red line represents a retrieved profile using Tikhonov regularization, the blue represents the retrieved profile using OEM; (c) the total column averaging kernel of CFC-12.

## 2.5 Error analysis

We analyze the smoothing error, forward model error, model parameter error and measurement error of the target gases based on the posteriori error estimation method described in Rogers (Rodgers and Connor, 2003). The error items and their relative uncertainties in the error budget are listed in Table 3. For the uncertainty of atmospheric temperature, the systematic error is about 2 K for the vertical range from 0 to 30 km, 5-9 K above 30 km, and the temperature random error is 5 K for the whole



atmosphere. The systematic and random uncertainties of solar zenith angle (SZA) are 0.1° and 0.2°, respectively. The line intensity uncertainty of CFC-11 and CFC-12 are 7% and 1%, respectively, referring to the maximum absorption coefficient

200 error given in pseudo-line-lists. In the error budget estimation of CFC-11, zero level offset (zshift) is included in the retrieval parameters error.

The total errors for CFC-11 and CFC-12 are about 4.12% and 1.79%, respectively, based on the combination of random and systematic errors. The systematic error and random error for CFC-11 are 4.07% and 0.66%, respectively. The line intensity and $H_2O$ spectroscopy in CFC-11 are the dominating systematic errors, with errors of 2.88% and 2.87%, respectively.

205 Temperature error is the dominating random error for CFC-11. For CFC-12, the systematic error and random error are 1.32% and 1.21%, respectively, while the dominating errors are temperature, $H_2O$ spectroscopy and zshift. The systematic error for CFC-11 is 7.61% and the random error is 3.08%, and for CFC-12, the systematic error is 2.24% and the random error is 2.40% at the St. Petersburg site (Polyakov et al., 2021). Our error estimates are reasonable.

210 **Table 3. Random and systematic error uncertainty and budget for CFC-11 and CFC-12 retrieval.**

| Error source | CFC-11 | | | CFC-12 | | |
|---|---|---|---|---|---|---|
| | Uncertainty/ % | Systematic/ % | Random/ % | Uncertainty/ % | Systematic/ % | Random/ % |
| Smoothing error | - | 0.04 | - | - | 0.02 | - |
| Measurement error | - | - | 0.33 | - | - | 0.10 |
| Retrieval parameters | - | 0.16 | | - | 0 | |
| Interfering species | - | 0.02 | | - | 0.01 | |
| Temperature | - | 0.08 | 0.52 | - | 0.20 | 0.84 |
| SZA | 0.1(0.2) | 0.09 | 0.18 | 0.1(0.2) | 0.23 | 0.11 |
| Line intensity | 7 | 2.88 | - | 1 | 0.27 | - |
| Temperature dependence of line width | 7 | 0.001 | - | 1 | 0.56 | - |



| Air-broadening of line width | 7 | 0.01 | - | 1 | 0.27 | - |
|---|---|---|---|---|---|---|
| H$_2$O spectroscopy | 10 | 2.87 | - | 10 | 0.67 | - |
| ILS | 2 | 0.01 | 0.01 | 2 | 0.12 | 0.12 |
| zshift | - | - | - | 1 | 0.85 | 0.85 |
| Total | - | 4.07 | 0.66 | - | 1.32 | 1.21 |

## 3 Results and discussion

### 3.1 Time series of CFC-11 and CFC-12 at the Hefei site

Figure 3(a) shows the time series of the CFC-11 total columns observed from January 2017 to December 2020 at the Hefei site. Figure 3(b) shows CFC-12 total columns observed from September 2015 to December 2020 at the Hefei site. The average total column of CFC-11 and CFC-12 is $(4.65 \pm 0.18) \times 10^{15}$ molec·cm$^{-2}$, and $(1.04 \pm 0.02) \times 10^{16}$ molec·cm$^{-2}$, respectively. The time series are fitted by a lowpass filtered fast Fourier transform (FFT) technology and a linear fitting to simulate the seasonal and interannual variation of CFC-11 and CFC-12 (Thoning et al., 1989). CFC-11 and CFC-12 show an obvious seasonal variation and annual decreasing trend. The annual decline of CFC-11 and CFC-12 is due to the prohibition of emissions from industrial production. For CFC-11, the annual decreasing rate of total column is $(-0.47 \pm 0.16)$ %/y$^{-1}$, which is close to the value of $-0.40$ %/y$^{-1}$ at St Petersburg observed from 2009 to 2019, but lower than the value of $(-0.86 \pm 0.12)$ %/y$^{-1}$ reported at St Denis and Maïdo station observed from 2004 to 2016, and $(-0.79 \pm 0.06)$ %/y$^{-1}$ derived from ACE-FTS during from 2012 to 2018 covering the region between 30°S and 30°N (Steffen et al., 2019; Polyakov et al., 2021; Zhou et al., 2016). For CFC-12, the annual decreasing rate of the total column is $(-0.79 \pm 0.31)$ %/y$^{-1}$ at the Hefei site, and close to the value of $(-0.76 \pm 0.05)$ %/y$^{-1}$ derived from St Denis and Maïdo measurements, and $(-0.79 \pm 0.06)$ % /y$^{-1}$ from ACE-FTS observations between 30°S and 30°N latitude, but larger than the value of $-0.49$%/y$^{-1}$ from St. Petersburg measurements (De Maziere et al., 2018; Polyakov et al., 2021; Steffen et al., 2019). The total column decline rate of CFC-11 is significantly lower than that of CFC-12.



(a)

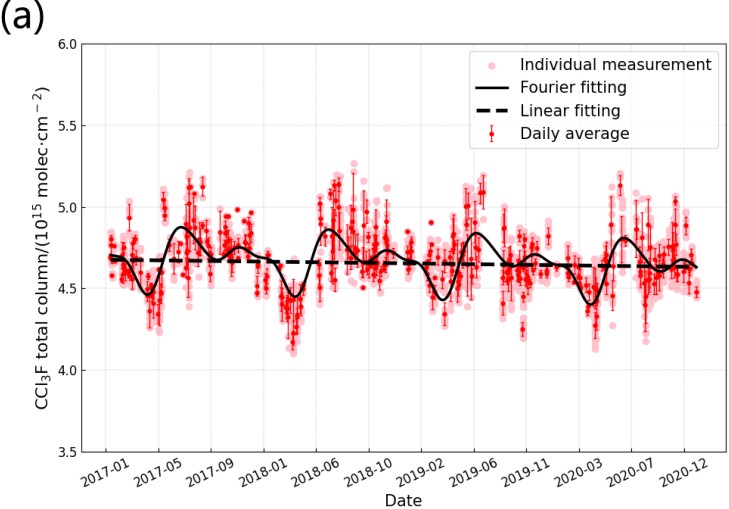

(b)

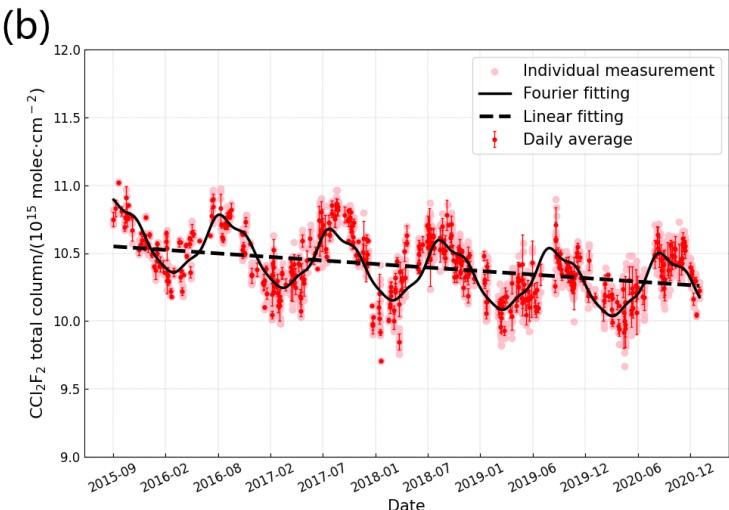

**Figure 3: The time series of the total columns of (a) CFC-11 and (b) CFC-12 from FTIR measurements at Hefei. The light red dots are the individual measurements, the red dots are the daily average, the error bars are standard deviations of the daily average, the black solid line and the black dash line are the fitting curve of individual measurements and the linear fitting curve, respectively.**

Compared to the total column, the near-surface concentration of the target gas can directly reflect the impact of local anthropogenic emissions. The CFC-11 and CFC-12 near-surface (at about 200 m height) VMR (volume mixing ratio) over Hefei are given in Fig. 4. The annual decreasing rate of near-surface VMR is (−0.60 ± 0.26) %/y$^{-1}$ for CFC-11, and (−0.81 ± 0.25) %/y$^{-1}$ for CFC-12. It can be seen that the annual decreasing rate of near-surface CFC-11 is higher than that of the total column, while the difference between the annual decreasing rate of near-surface CFC-12 and the



total column is relatively small. The difference between the annual decreasing rate of near-surface CFC-11 and column may be due to the fact that long-distance transmission of airmass affect more total column than near-surface concentration.

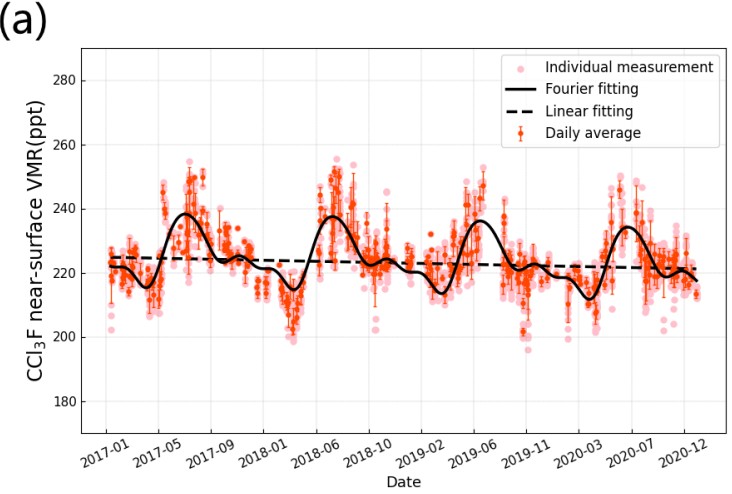

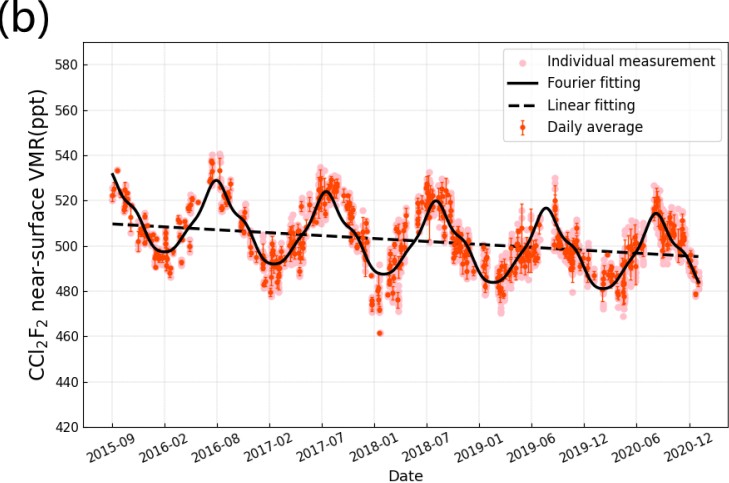

**Figure 4: The time series of the near-surface VMR of (a) CFC-11 and (b) CFC-12 from FTIR measurements at Hefei. The light red dots are the individual measurements, the orange dots are the daily average, the error bars are standard deviations of the daily average, the black solid line and the black dash line are the fitting curve of individual measurements and the linear fitting curve, respectively.**

The seasonal variations of de-trended CFC-11 and CFC-12 total columns are given in Fig. 5, and the seasonal variations of

near-surface VMR are given in Fig. 6, respectively. The de-trended values are obtained by subtracting the annual average long-term trends from individual measurements at the Hefei site. Both CFC-11 and CFC-12 show an obvious seasonal variation. For total column, CFC-11 has the highest concentration in summer and a trough in spring, and CFC-12 has the highest





concentration in summer and autumn and a trough in spring. The peak value of CFC-11 appears in July and the minimum appears in April, with a seasonal amplitude of $3.89 \times 10^{14}$ molec·cm$^{-2}$ and a seasonal variability of 8%. The peak of CFC-12 is

in September and the minimum is in March, with a seasonal amplitude of $4.53 \times 10^{14}$ molec·cm$^{-2}$ and a seasonal variability of 4%. Compared with CFC-12, CFC-11 has small difference in autumn and winter. For near-surface concentration, the peak value of CFC-11 appears in July and the minimum appears in April, with a seasonal amplitude of 21 ppt and, a seasonal variability of 9%. The peak of CFC-12 is in August and the minimum is in February, with a seasonal amplitude of 32 ppt and, a seasonal variability of 6%. The monthly variation of CFC-11 total column and near surface VMR are consistent, whereas

CFC-12 total column variations have one-month phase delay relative to near surface VMR. Near-surface measurements are more affected by local emissions than total column measurements, so the near-surface concentration variations reflect the variations of local emissions. The seasonal variation of the total column is also affected by emissions from distant sources in the upper atmosphere, and this cause the phase delay between the total column and the near-surface concentration (Te et al., 2016). In addition, more use of air conditioning and other refrigeration equipment in summer, and foams releasing more CFCs

at high temperatures lead to high concentrations of atmospheric CFCs (Wan et al., 2009). Yang et al. (2021) measured a higher CFCs concentration in August at the top of Mount Tai in northern China from June 2017 to April 2018 (Yang et al., 2021). CFC-11 and CFC-12 at St Denis and Maïdo stations also show a seasonal cycle with high concentrations in summer and autumn (Zhou et al., 2016).

The utilizations of CFC-11 and CFC-12 are not exactly the same in China. CFC-11 is often used as blowing agent and tobacco

shred expander, and CFC-12 is mostly used as refrigerant and blowing agent (Wang et al., 2010). The difference between their emission sources may explain the difference between their near-surface seasonal variations. The inconsistency in CFC-12 and CFC-12 lifetime may be the other reason for their seasonal variations difference. Primary sinks of CFC-11 and CFC-12 are in stratosphere, and the lifetime of CFC-11 is shorter than CFC-12, so CFC-11 has more depletion with height in the stratosphere due to photochemical destruction. Nevison et al. (2004) proposed that CFC-11 has a shorter lifetime than CFC-12 and greater

sensitivity to stratospheric downwelling, which causes the greater seasonal variability in CFC-11 (Nevison et al., 2004).

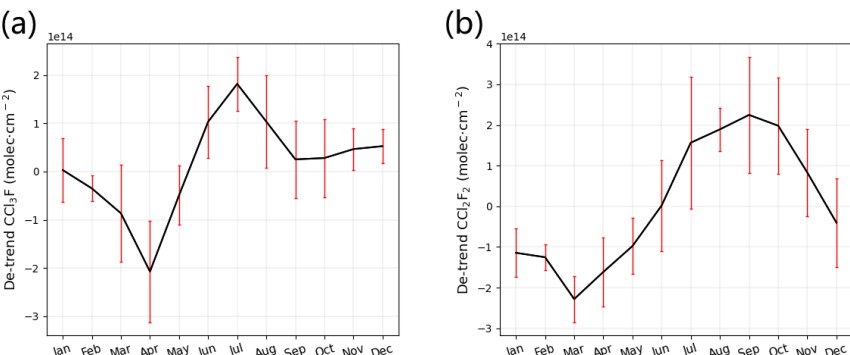

**Figure 5: (a) The de-trended total columns of CFC-11; (b) the de-trended total columns of CFC-12. The error bars show the standard deviation of monthly averaged value.**





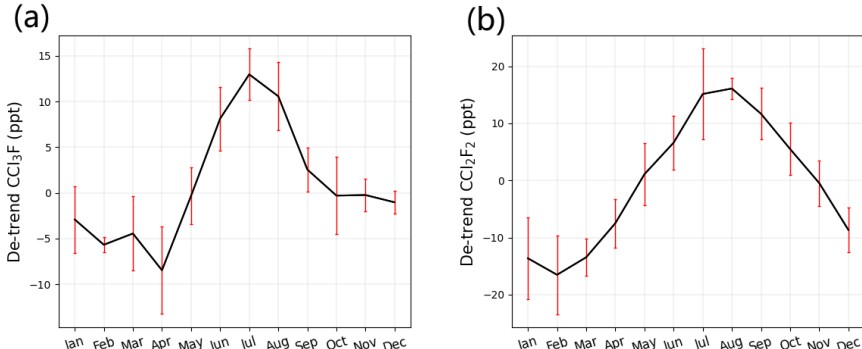

Figure 6: (a) The de-trended near-surface VMR of CFC-11; (b) the de-trended near-surface VMR of CFC-12. The error bars show the standard deviation of monthly averaged value.

## 3.2 Comparison with satellite and model data

The Atmospheric Chemistry Experiment Fourier Transform Spectrometer (ACE-FTS) was launched onboard the SCISAT-1 satellite and recorded 16 halogen-containing gases, which provide the data for study of ozone chemistry and dynamics processes in the stratosphere and upper troposphere (Bernath, 2002; Boone et al., 2004; Bernath, 2017). The ACE-FTS is a high-resolution (0.02 cm$^{-1}$) spectrometer with a spectral range of 750–4400 cm$^{-1}$, which operates in the solar occultation mode, and continuously collects infrared solar spectra from 150 km altitude down to the cloud top (Mahieu et al., 2008). ACE-FTS observations cover almost the whole world, but the main observing target is not China. ACE is mainly aimed at study of ozone chemical process at high latitude, so there are a few observations for tropical and subtropical areas. We choose the satellite data centered at the Hefei site with latitude of ±5° and longitude of ±10° (27°N–37°N, 107°E–127°E). In this study, we use the v4.1/v4.2 Level 2 ACE data. The observation period is from 2017 to 2020 for CFC-11 and from 2015 to 2020 for CFC-12. The method of Brown et al. (2011) was adopted to eliminate the points deviating from the 2.5 times median absolute deviations (MAD) to filter the outliers (Brown et al., 2011). The a priori profile and vertical sensitivity of ACE-FTS and ground-based FTIR are different, so it is difficult to directly compare the raw profiles observed from ACE-FTS with FTIR data. In order to compare the two data, we interpolated the profiles of ACE-FTS to the FTIR vertical grid, and smoothed the interpolated data by the FTIR averaging kernel and a priori profile using the method of Rodgers and Connor (2003) (Rodgers and Connor, 2003), that is:

$$x_{\text{smooth}} = x_a + \mathbf{A}(x_{\text{sat}} - x_a) \tag{4}$$

where $x_a$ and $\mathbf{A}$ are the a priori profile and averaging kernel of FTIR observations, respectively, and $x_{\text{sat}}$ is the satellite profile after interpolation, $x_{\text{smooth}}$ is the smoothed profile of the satellite. The profile obtained from satellite and the ground-based FTIR are shown in Fig. 7. The vertical VMR profiles of CFC-11 measured by ground-based FTIR are slightly larger than ACE-FTS VMR profiles below 30.5 km. The measured CFC-12 profiles by FTIR are slightly smaller than ACE-FTS profiles below 20.5 km.



The dry-air averaged mole fraction ($X_{gas}$) of the target gas from ACE-FTS and FTIR is calculated as follows:

$$X_{gas} = \frac{cloumn_G}{cloumn_{dry\ air}} \tag{5}$$

where $cloumn_G$ and $cloumn_{dry\ air}$ are columns obtained from target gas and dry air, respectively. The retrieved profiles of CFC-11 for ACE-FTS are mainly at 5-23 km, and CFC-11 and CFC-12 are mainly distributed below 20 km. So we refer to the study of Steffen et.al (Steffen et al., 2019), and calculate $X_{gas}$ of CFC-11 and CFC-12 at 5.5-17.5 km. The dry-air averaged mole fractions of CFC-11 and CFC-12 obtained from the ACE-FTS satellite are $(221 \pm 4)$ ppt and $(527 \pm 13)$ ppt, respectively, 

while the column-averaged dry air mole fractions from FTIR observation are $(232 \pm 11)$ ppt and $(501 \pm 14)$ ppt, respectively. The relative difference of ACE and FTIR data is calculated by concentration from satellite minus FTIR divided by FTIR data at the same altitude. The mean relative difference between the two data is $(-5.6 \pm 3.3)$ % and $(4.8 \pm 0.9)$ % for CFC-11 and CFC-12 profiles from 5.5 to 17.5 km, respectively. The results demonstrate our FTIR data agree relatively well with the ACE-FTS satellite data.

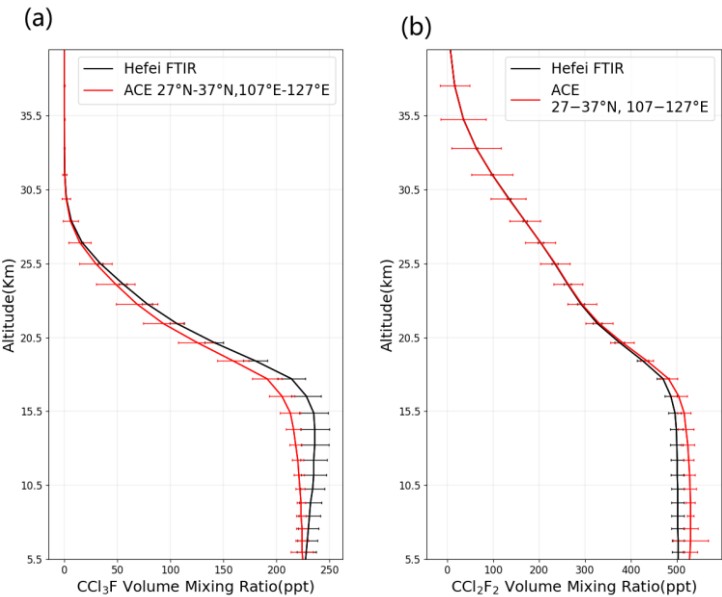


**Figure 7: The vertical profile of (a) CFC-11 and (b) CFC-12 obtained from ground-based FTIR (black) and ACE-FTS satellite (red) measurements. The error bars are the standard deviations of the mixing ratio profile for CFC-11 and CFC-12 at each layer.**

The spatial range of satellite data selected is wide, the observation time is not the same as FTIR observations, and the priori profiles of satellite and FTIR data are different, leading to the difference between ACE-FTS and FTIR data. Mahieu et al. 

(2008) compared the profiles retrieved from the balloon-borne FTIR spectrometer MkIV with ACE-FTS data, and found that the VMR concentrations of ACE-FTS were systematically smaller than MKIV values for CFC-11, with a difference of $-10\%$ above 12 km and about $-20\%$ below 12 km, while for CFC-12, ACE-FTS VMR concentrations are systematically lower than



MkIV values, with maximum differences of −10% (Mahieu et al., 2008). For the in-situ measurements of CFCs in China, Yang et al. (2021) reported that the concentrations of CFC-11 and CFC-12 measured at the top of Mount Tai in northern China from
2017 to 2018 were 257 ppt and 577 ppt, respectively (Yang et al., 2021). The surface mean mixing ratio of CFC-11 in five cities in China (Beijing, Hangzhou, Guangzhou, Lanzhou and Chengdu) observed from 2017 to 2019 was in the range of 244 to 268 ppt, and that of CFC-12 ranged from 526 to 585 ppt during 2015 to 2019 (Yi et al., 2021). Benish et al. (2021) found concentrations of (281 ± 44) ppt for CFC-11 and (552 ± 93) ppt for CFC-12 from aircraft observations at 500-3500 m above Hebei Province, China in 2016 (Benish et al., 2021). The reported values observed from different locations in China are similar
to the dry-air averaged mole fraction of CFC-11 and CFC-12 measured at Hefei, which reflects the reliability of our results. On the other hand, the lower concentrations of CFC-11 and CFC-12 may be due to the smaller emissions at Hefei.

Table 4 lists the annual decreasing rate of CFC-11 and CFC-12 at the Hefei site calculated from ground-based FTIR data, ACE-FTS satellite data and the data from WACCM V6. WACCM V6 data are available from the website ftp://nitrogen.acom.ucar.edu/user/jamesw/IRWG/2013/WACCM/V6/ (last access: 20 January 2022). The annual decreasing
rate of CFC-11 obtained from FTIR total column, FTIR near-surface VMR, ACE-FTS and WACCM is (−0.47 ± 0.16) %, (−0.60 ± 0.26) %, (−1.15 ± 0.22) %, and (−1.68 ± 0.18) %, respectively. ACE-FTS and WACCM significantly overestimates the decreasing trend of CFC-11. The annual decreasing rate of CFC-12 from FTIR total column, FTIR near-surface VMR, ACE-FTS and WACCM data is (−0.79 ± 0.31) %, (−0.81 ± 0.25) %, (−0.85 ± 0.15) % and (−0.81 ± 0.05) %, respectively. The three independent values are very close. The wide observation range of ACE-FTS and few matching data for Hefei observations
lead to low representativeness of the annual rate for ACE-FTS data. Polyakov et al. (2021) also found the decreasing trend of FTIR data is different from ACE-FTS data and WACCM data, and the difference of CFC-11 is greater than that of CFC-12 (Polyakov et al., 2021).

**Table 4: Summary of annual decreasing rate obtained from measurements of FTIR, ACE-FTS satellite and WACCM data**

|  | FTIR (total column) | FTIR (near-surface) | ACE-FTS (8.5-17.5 km) | WACCM |
|---|---|---|---|---|
| CFC-11 | −0.47 ± 0.16 % | −0.60 ± 0.26 % | −1.15 ± 0.22 % | −1.68 ± 0.18 % |
| CFC-12 | −0.79 ± 0.31% | −0.81 ± 0.25 % | −0.85 ± 0.15 % | −0.81 ± 0.05 % |

**3.3 Comparison with data from other FTIR site**

Further, we compared our data with those from other NDACC station. NDACC is an international global atmospheric observation network, which operates a variety of high precision ground-based observation technologies, and provides long-term observations of a variety of atmospheric components (De Maziere et al., 2018). The NDACC data used were obtained by the ground-based high-resolution FTIR instrument at St. Petersburg station, from the NDACC database (https://www-
air.larc.nasa.gov/missions/ndacc/data.html/, last access: 02 March 2022). The CFC-11 data cover the observation period from



January 2017 to March 2019, and CFC-12 from September 2015 to December 2020. This station is located at the latitude and longitude of 59.9°N and 29.8°E, and the observing instrument and the retrieval spectral window of CFC-11 used are the same as those at Hefei site, and St. Petersburg station CFC-12 retrieval spectral window is 1160-1162 cm⁻².

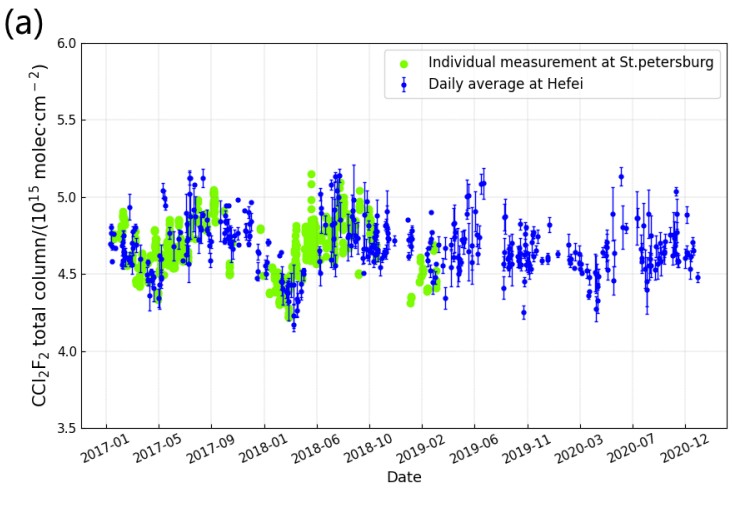

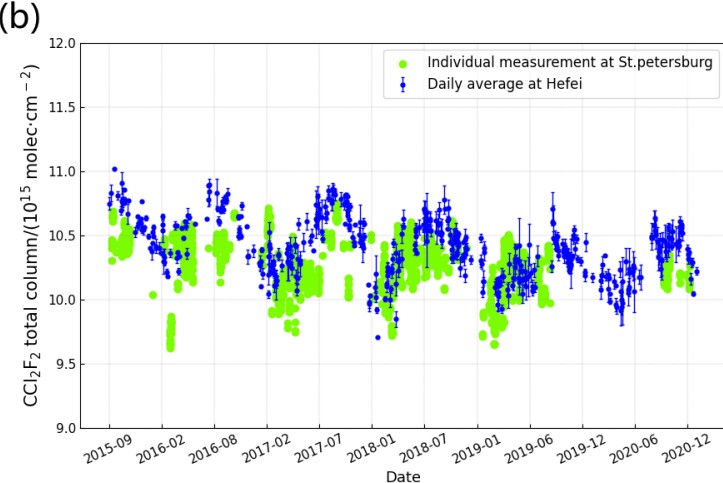

**Figure 8: The time series of the total columns of (a) CFC-11 and (b) CFC-12 from Hefei and St. Petersburg FTIR measurements. The dark blue dots are the daily average at Hefei, the error bars are the standard deviations of the daily average at Hefei, the green dots are the individual measurements at St. Petersburg.**

The total columns of CFC-11 at the Hefei site are very close to those at St. Petersburg station, while the total columns of CFC-12 at Hefei is slightly higher than those at St. Petersburg station, as seen in Fig. 8. The mean difference of total columns of CFC-11 between the two data sets is $3.63 \times 10^{12}$ molec·cm⁻², while the mean difference of total columns of CFC-12 is $1.69 \times 10^{14}$ molec·cm⁻². In the study of Polyakov et al. (2021), the annual decreasing rate of CFC-11 and CFC-12 is −0.40% yr⁻¹ and −0.49% yr⁻¹ from 2009 to 2019 at the St. Petersburg site, respectively (Polyakov et al., 2021). The annual decreasing




rate of CFC-11 and CFC-12 at Hefei is greater than that the corresponding value at St. Petersburg. The correlation coefficient
(R) between the monthly averaged total column observed at Hefei and St. Petersburg for CFC-11 and CFC-12 is 0.59 and 0.60
(Fig. 9). In addition, the seasonal variation of CFC-11 and CFC-12 at Hefei is slightly different from that at St. Petersburg. The
two target gases at St. Petersburg have a maximum in autumn, showing a phase shift in seasonal variation for the Hefei site.
The two sites are located at different latitude, which may explain the different seasonal variation of the two gases. Also, the
seasonal variation of the CFCs at St. Petersburg are mainly due to the variation of surface pressure, water vapor and emissions
from anthropogenic pollution sources (Polyakov et al., 2021). However, the seasonal variation in CFCs in China may be mainly
caused by variations in emissions from CFC sources, and the anthropogenic emissions are greater in summer (Yang et al., 2021;
Zhang et al., 2011).

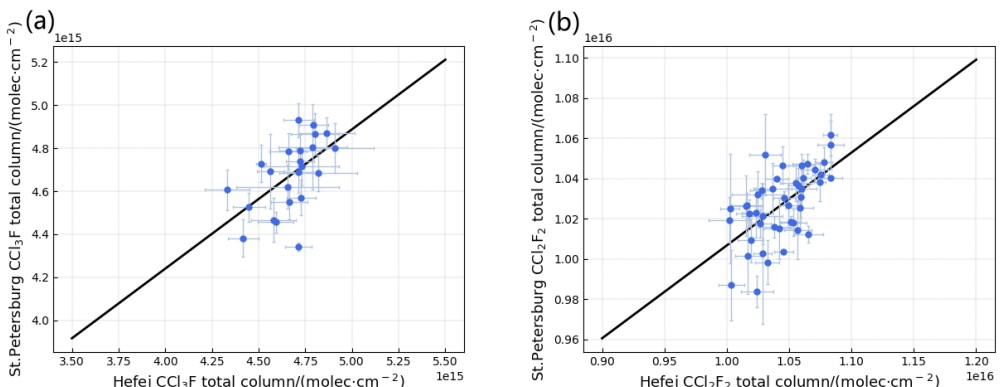

**Figure 9: Correlation plot of the coincident monthly averaged total column of (a) CFC-11 (b) CFC-12 from Hefei and St.**
**Petersburg measurements. Black line is the linear regression curve between Hefei and St. Petersburg data.**

**4 Conclusion**

In this study, the atmospheric CFC-11 and CFC-12 vertical profiles and total columns are retrieved from ground-based FTIR
measurements over Hefei, China, during January 2017 to December 2020, and September 2015 to December 2020, respectively.
The seasonal variation and annual trend of the two gases are analyzed, and then the data are compared with other independent
datasets, such as satellite data and model simulations. This is the few report about the detection of CFC-11 and CFC-12 columns
and their tempo-spatial variations in China.

Tikhonov $\mathbf{L_1}$ regularization was applied to constrain the retrieved profile in the retrieval strategy. Atmospheric CFC-11 and
CFC-12 are mainly distributed within 0-20 km vertical atmosphere. The total retrieval error is 4.12% for CFC-11, and 1.79%
for CFC-12. CFC-11 and CFC-12 total columns over Hefei showed a decreasing rate of (−0.47 ± 0.16) % per year and (−0.79
± 0.31) % per year, respectively. The near-surface VMR of CFC-11 and CFC-12 over Hefei showed a decreasing rate of (−0.60
± 0.26) % per year and (−0.81 ± 0.25) % per year, respectively. CFC-11 total columns are higher in summer, and CFC-12 total
columns are higher in summer and autumn. Both of CFC-11 and CFC-12 total columns are lower in spring. The seasonal
amplitude between the maximum value of CFC-11 in July and the minimum value in April is 3.89×10$^{14}$ molec·cm$^{-2}$, while



CFC-12 has the peak in September and the minimum in March, with a difference of $4.53 \times 10^{14}$ molec·cm$^{-2}$. The near-surface
CFC-11 concentration appears the maximum in July and the minimum in April, with a seasonal amplitude of 21 ppt, and CFC-12 has the maximum in August and the minimum in February, with a seasonal amplitude of 32 ppt.

Further, we compared FTIR data with the ACE-FTS satellite and WACCM data, as well as the data from other NDACC station. The dry-air averaged mole fractions of CFC-11 and CFC-12 calculated from the altitude of 5.5 to 17.5 km for ACE-FTS satellite data is $(221 \pm 4)$ ppt and $(527 \pm 13)$ ppt, while the column-averaged dry air mole fractions from FTIR observations
are $(232 \pm 11)$ ppt and $(501 \pm 14)$ ppt, respectively. The mean relative difference between the FTIR and ACE-FTS concentrations at the altitude from 5.5 to 17.5 km is $(-5.6 \pm 3.3)$ % and $(4.8 \pm 0.9)$ % for CFC-11 and CFC-12, respectively. The results demonstrate our FTIR data agree relatively well with the ACE-FTS satellite data. Then the interannual variations from ground-based FTIR measurements, ACE-FTS observations and WACCM V6 data for CFC-11 and CFC-12 were compared. The annual decreasing rate of CFC-11 measured from ACE-FTS and calculated by WACCM V6 are $(-1.15 \pm 0.22)$ %
and $(-1.68 \pm 0.18)$ %, respectively. ACE-FTS and WACCM data clearly overestimated the decreasing rate, the corresponding value of FTIR total column and near-surface data is $(-0.47 \pm 0.16)$ % and $(-0.60 \pm 0.26)$ %, respectively. The annual decreasing rate of CFC-12 from ACE-FTS and WACCM V6 is $(-0.85 \pm 0.15)$ % and $(-0.81 \pm 0.05)$ %, respectively, which are close to the corresponding value $(-0.79 \pm 0.31)$ % from the FTIR total column measurements and $(-0.81 \pm 0.25)$ % from the FTIR near-surface data. The total columns of CFC-11 at Hefei are very close to those at St. Petersburg station, with a mean difference
of $3.63 \times 10^{12}$ molec·cm$^{-2}$, while CFC-12 is slightly higher at Hefei, with the mean difference of $1.69 \times 10^{14}$ molec·cm$^{-2}$. The correlation coefficient (R) between the monthly averaged total column observed at Hefei and St. Petersburg for CFC-11 and CFC-12 is 0.59 and 0.60, respectively. The differences between the CFCs columns at the two sites are due to the different CFCs emission sources and different latitude.

*Data availability.* The FTIR CFC-11 and CFC-12 retrievals at Hefei are available by contacting the corresponding author.
ACE-FTS data are publicly available via the https://databace.scisat.ca/level2/ (last access: 25 January 2022). WACCM V6 data are available from the website ftp://nitrogen.acom.ucar.edu/user/jamesw/IRWG/2013/WACCM/V6/ (last access: 20 January 2022). The FTIR CFC-11, CFC-12 retrievals at the St. Petersburg site are available from https://www-air.larc.nasa.gov/missions/ndacc/data.html/ (last access: 02 March 2022).

*Author contributions.* XyZ retrieved the data and wrote the manuscript. WW and CgS designed the experiment. CL and
YX contributed to the discussion of the paper and explaination the results, PW and QqZ took part in FTIR measurement, and AlP provided retrieval guidance and St. Petersburg data.



*Competing interests.* The authors have the following competing interests: At least one of the coauthors is a member of the editorial board of Atmospheric Measurement Techniques. The peer-review process was guided by an independent editor, and the authors have also no other competing interests to declare.

*Acknowledgements.* We gratefully acknowledge the support of the National Key Technology R&D Program of China (2019YFC0214702), the National Natural Science Foundation of China (41775025), the Major Projects of High Resolution Earth Observation Systems of National Science and Technology (05-Y30B01-9001-19/20-3), the Strategic Priority Research Program of the Chinese Academy of Sciences (XDA23020301), the National Key Project for Causes and Control of Heavy Air Pollution (DQGG0102 and DQGG0205), the Natural Science Foundation of Guangdong Province (2016A030310115), and State Environmental Protection Key Laboratory of Sources and Control of Air Pollution Complex (No. SCAPC202110). Thanks Professor Nicholas Jones, School of chemistry, Wollongong University, Australia, for guidance on ground-based spectroscopy retrieval.

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
