# Peer review of "Retrieval of Atmospheric CFC-11 and CFC-12 from Highresolution FTIR Observations at Hefei and Comparisons with other Independent Datasets"

_EGUsphere, 2022_

## Author Comment (AC1)

**Reviewer #1**

The manuscript "Retrieval of Atmospheric CFC-11 and CFC-12 from High resolution FTIR Observations at Hefei and Comparisons with Satellite Data" by Zeng et al. describes the retrieval of atmospheric ozone depleting chlorofluorocarbons CFC-11 and CFC-12 from solar absorption spectra measured using a high resolution Fourier Transform infrared spectrometer at Hefei, China, and examines the resulting, multi-year timeseries, and compares it to satellite measurements and concentrations prognosed by a model.

The retrieval scheme builds on work carried out at the St Petersburg NDACC-IRWG (Network for the Detection of Atmospheric Composition Change – InfraRed Working Group) station. The novelty here is the retrieval of these species from the spectra measured at Hefei, representing one of the few measurements of its kind in China. The long-term monitoring of key atmospheric constituents such as these and the understanding of their evolution within a global context is important and the publication of these results should be encouraged.

The manuscript is generally well structured and written but would benefit from further development of several sections to provide more of a thorough description of some of the important concepts as described below under specific comments.

Subject to the incorporation of these changes and the corrections suggested under technical corrections below, publication of this manuscript is recommended.

Response: We appreciate your constructive and positive comments. The comments and proposed corrections have been taken into account and helped improving the paper. Each comment has been addressed as follows. There is an extensive discussion among the authors regarding how to revise the content. Line numbers refer to the revised manuscript (version without tracked changes).

**Specific comments**

The manuscript presents retrievals of CFC-11 between January 2017 and December 2020 and CFC-12 between September 2015 and December 2020. The authors should explain why the two observing periods are different.

Response: The spectral range for retrieval of CFC-11 is 830-860 cm-1, while the spectral range for retrieval of CFC-12 is 1160.2–1161.4 cm-1. We replaced CaF2 incoming light window with KCl window for FTIR spectrometer in December 2016, which increased the covering spectral range from greater than 1000 cm-1 to greater than 700 cm-1. So we can retrieve CFC-11 since then. The explanation is included on Line 128-130 in Section 2.1.

The abstract states that comparisons are made to other NDACC stations. It should be made clearer whether Hefei is or is not an NDACC-IRWG station. Also make it clear that the comparison is with NDACC-IRWG stations, not other NDACC observations.

Response: Hefei is not an NDACC-IRWG station now, but is applying to join the NDACC-IRWG. We added this explanation on Line 121-122 in Section 2.1. We also make it clear that the comparison is

**with NDACC-IRWG stations, not other NDACC observations.**

The abstract also introduces the comparison to ACE-FTS satellite measurements and WACCMv6 model and presents quotative results of the comparison. It would provide important context here to define the spatial extent of the satellite/model data used i.e., global or coincident with Hefei.

Response: The spatial extent of ACE-FTS satellite data coincident with Hefei is centered at the Hefei site with latitude of  $\pm 5^{\circ}$  and longitude of  $\pm 10^{\circ}$ . The WACCMv6 simulated data consider the Hefei site (31.9°N, 117.17°E) as the center, with a horizontal resolution of  $0.95^{\circ} \times 1.25^{\circ}$ . We added this explanation on Line 341 in Section 3.2.

In its current form, Section 2.2. does not provide sufficient information to allow the reader to reproduce the author's results. For example, how was the pseudo line list produced and how can it be obtained? Also, Table 1 lists zshift and beam as background retrieval parameters for CFC-11, but these are not described or explained in the text.

Response: The address providing the pseudo-line-lists is included on Line 143 in Section 2.2. Also, we added the description about zshift and beam correction on Line 147-152 in Section 2.2.

In Section 2.3, How is the measurement error used to refine the regularization strength determined? This may be covered in the cited article, but it is probably important enough to discuss within the manuscript.

Response: We modified the content of this part. According to Steck (2002), we can determine the unique value of  $\alpha$  for the given mean retrieval measurement noise error  $\sigma_m$ , the measurement noise error can be selected by interest, but should be considered within a reasonable range (20% in citation). The smoothing error can also be considered to determine the value of  $\alpha$ . Steck (2002) indicated that we can consider these two errors at the same time, and the method is to minimize the total error calculated by measurement noise error and smoothing error (because the dependence of the forward model parameter error on  $\alpha$  is equivalent to the measurement noise error, the consideration of the forward model error is ignored). According to the posteriori error estimation method, we calculate the total error of the measurement noise error and the smoothing error  $\mathbf{S}_{tot} = \sqrt{\mathbf{S}_m^2 + \mathbf{S}_s^2}$ , and the results are list on Line 181-191 in Section 2.3.

It is unusual to see a column averaging kernel that contains as much structure and sharp transitions as the ones plotted in panel c of Figs 1 and 2. It would be good to include the layer averaging kernels, or a subset thereof, to see how this has come about.

Response: Following the suggestions from the two reviewers, we replaced the total column averaging kernels with layer averaging kernels in Fig. 2(c) and 3(c).

Not all sources of error listed in Table 3 are mentioned in the text of Section 2.5. These sources and the associated assumptions concerning their magnitude should be discussed.

Response: We added the other missing sources of error and their magnitude on Line 221 and 223 in Section 2.5, including the uncertainty of temperature dependency of line width, air widening of line width, H2O spectroscopy and ILS.

At P10 L215, the sentence "The time series are fitted by a lowpass filtered fast Fourier transform (FFT) technology and a linear fitting to simulate the seasonal and interannual variation of CFC-11 and CFC-12 (Thoning et al., 1989)" may not accurately describe the timeseries decomposition process. It appears from Fig. 3 that a linear trend and multi-harmonic seasonal cycle have been fitted. The authors should consider revising this statement and state the number of harmonic terms that have been used to fit the seasonality.

Response: We used the second-order polynomial and the four harmonic terms to fit the seasonal periods of CFC-11 and CFC-12, where t is the time fraction in years, and the equation F(t):

$$F(t) = a + b \cdot t + c \cdot t^{2} + \sum_{k=1}^{4} \left( d_{2k-1} \cos \left( 2\pi kt \right) + d_{2k} \sin \left( 2\pi kt \right) \right)$$

where a is the intercept, b and c represent polynomial fitting term coefficient, and  $d_1$  to  $d_8$  represent sin/cosine harmonic term coefficient.

In the revised manuscript, we modified the fitting according to other sites, and reused first-order polynomial and three harmonic terms to simulate CFC-11 and CFC-12, the equation F(t) become:

$$F(t) = a + b \cdot t + \sum_{k=1}^{3} \left( c_{2k-1} \cos \left( 2\pi kt \right) + c_{2k} \sin \left( 2\pi kt \right) \right)$$

where *a* is the intercept, *b* represent annual trend, and  $c_1$  to  $c_6$  represent sin/cosine harmonic term coefficient. The annual trend is also obtained by the new fitting formula *F*(t). We revised the statement, and describe the fitting formula in detail on Line 239-243 in section 3.1.

In section 3.1 two retrieval products are discussed: the total columns and near surface concentrations. These products should be introduced prior to their discussion. It would be helpful to do this as part of a discussion of the information content of the retrieval process possibly as its own sub-section in section 2. The error analysis should also state how the retrieval errors propagate into these two products.

Response: Following the suggestion of the reviewer 2, the discussion part about near surface concentrations was deleted, due to the insufficient sensitivity. The error analysis for retrieval of total columns is discussed in Section 2, and we added the error calculation formula in detail on Line 169-171 in Section 2.3.

In the conclusions, the statement that "ACE-FTS and WACCM data clearly overestimated the decreasing rate,.." doesn't appear to be justified in the context of the evidence presented given the spatio-temporal differences between the measurements. This should be revised.

Response: We modified the statement to "The decreasing trend of ACE-FTS and WACCM is significantly higher, while the corresponding value of FTIR total column is  $-0.47 \pm 0.06$  % yr-1".

It would be good to see some stronger conclusions drawn, for example placing the findings of this work in the context of previously published findings and a comment on the differing types of emissions that lead to the difference between Hefei and St. Petersburg.

Response: There were few reports on CFC in Hefei and St. Petersburg before. St. Petersburg is close to

the industrially developed European part of Russia, and there are some sources of CFC-11 and CFC-12. At the same time, the temperature in St. Petersburg is mild in summer (about 17 °C), and the temperature in Hefei is about 29 °C in summer, the use of air conditioners in summer should be lower than in Hefei. Some studies in China indicate that the leakage of CFCs caused by waste treatment in municipal solid waste landfills and low leak tightness of automobile mobile air conditioning systems on hot and humid days may be the potential source of CFCs (Zhen et al., 2020; Zhang et al., 2017). Section 3.3 has modified to compare with more NDACC sites, as for St. Petersburg's research on emission types is insufficient, so we will not discuss emission at St. Petersburg.

Are there any plans to continue or update the dataset? It would be good to include this information.

Response: We will update our dataset later. At present, Hefei site is applying to join NDACC-IRWG. When we join it successfully, we will upload our data to the NDACC database and update the data. We added this description to the Data availability of the revised manuscript.

**Technical corrections**

P1 L25 in abstract remove % sign after -0.47 to be consistent with the rest of the abstract, elsewhere when expressing a value and uncertainty the parentheses are unnecessary.

Response: We have modified the contents in the revised manuscript.

P2 L56 citation should be Montzka et al., 2021.

Response: We checked the citation, and this citation is Montzka et al., 2018. Montzka et al. (2018) in Nature Letter, wrote in the 4 paragraph of the paper shows "The gap between expectations and observations widened substantially after 2012, when CFC-11 global mole fractions began decreasing even more slowly. In recent data (from mid-2015 to mid-2017), the mean rate of change for CFC-11 ( $-1.0 \pm 0.2$  ppt yr-1, or  $-0.4 \pm 0.1\%$  yr-1) was about 50% slower than that observed during 2002–2012; it also was much slower than has been recently projected." The content is consistent with our citation, so there should be right here.

P2 L57 It might help the reader to know the type of atmospheric observations, in-situ or remote sensing.

Response: Observations in Gosan, South Korea, and Hateruma, Japan are in-situ observations. We added this explanation on Line 63 of the page 2.

P2 L60 insert a space between CFC-11 and and.

Response: We did it in the revised paper.

P2L60 check the units are correct for the emission rates (Gg not kg?) and use yr-1 to be consistent with the rest of the manuscript

Response: We corrected the unit for the emission rates.

P2 L64 "Study of the temporal-spatial distribution and variations of CFCs in the atmosphere is of great

significance to reduce stratospheric ozone depletion and greenhouse gas emissions." The study itself does not reduce the emissions, but it does improve understanding and suggest what needs to be done to facilitate reductions. Consider revising this sentence.

Response: We modified the statement to "Study of the temporal-spatial distribution and variations of CFCs in the atmosphere is of great significance for improving understanding and implementing policies to reduce stratospheric ozone depletion and greenhouse gas emissions".

P3 L80 this sentence may need a change of emphasis, in that HIRDLS, ILAS etc are not mainly used for CFC measurements, but they may be the main instruments used for this type of measurement.

Response: We changed the sentence to "Satellite remote sensing techniques, such as high resolution dynamics Limb Sounder (HIRDLS), Improved Limb Atmospheric Spectrometer (ILAS), the collocated Michelson Interferometer for Passive Atmospheric Sounding (MIPAS) and Atmospheric Chemistry Experiment Fourier transform spectrometer (ACE-FTS), also play an important role in measuring the global distribution of CFCs".

P3 L92 Throughout the manuscript there are sentences like this where the un-parenthesised citation is used at the beginning of the sentence with the parenthesised version at the end. It is unnecessary to include the citation twice in one sentence.

Response: We modified this kind of citation in the revised manuscript.

P5L 141 remove the word time and replace with either iteration or step, i.e., "iteration index i" or "step i".

Response: We modified it to "iteration index i".

P9 L207 Suggest starting the sentence introducing the error values from the Polyakov study with "At the St Petersburg site..." or similar, to avoid a little confusion.

Response: We modified the description here following this suggestion.

P9 L209 Last sentence should be elaborated.

Response: We modified this sentence to "Our error estimates are similar to those at the St. Petersburg station, and slightly smaller compared with the latter".

P9 Table 3, This table is a little hard to read, consider more use of horizontal lines to separate items

Response: We added horizontal lines to Table 3.

P10 L219 Throughout section 3.1 trends are given the units %/yr-1 when they should be %yr-1 (to be consistent with the rest of the manuscript) or %/yr

Response: We modified all units %/yr-1 to "% yr-1" in the manuscript.

P10 L225 Insert a space between -0.49 and %

Response: We did it in the manuscript.

Figures 3, 4 and 8: Consider using the same x axis scale for both timeseries to allow the reader to see the seasonal cycles aligned.

Response: Figures 4 use the same x axis scale for both timeseries. We deleted Figure 8, which shows timeseries comparison, but now added the comparison of monthly means of the two data in Figure 7 and 8.

P13 L254 In this discussion, are the seasonal amplitude and variability not the same? I.e., the amplitude in units of molecules per unit area or mixing ratio is also expressed as a percentage of annual mean or detrended mean?

Response: The seasonal amplitude and variability are not the same. We added the explanation of the seasonal amplitude and variability on Line 267-269 in Section 3.1.

P13 L256 consider starting a new paragraph to discuss near surface concentration seasonality

Response: The discussion about near surface concentration was deleted, according to the comments of reviewer 2, as the information content is not enough to obtain meaningful surface values from FTIR retrieval.

**P13 L271 CFC-11?**

Response: We modified it to "CFC-11".

P15 L304 It should be explained why the columns of dry-air mole fractions are being compared and not molecules per unit area that were discussed previously. Also, it is not apparent in the text how the dry-air column has been derived.

Response: Because the profile range measured by ACE-FTS satellite is inconsistent with the FTIR measurement range, the dry air mole fractions are calculated here for comparison. It is also facility to compare with the ground and upper air mixture ratios obtained from field measurements in other regions of China. We added the dry-air column formula in Eq. 8.

**P16L333 Are global WACCM data used or the same spatial criteria as the ACE-FTS? This should be made clear in the text.**

Response: the simulated data consider the Hefei site  $(31.9^{\circ}N, 117.17^{\circ}E)$  as the center, with a horizontal resolution of  $0.95^{\circ} \times 1.25^{\circ}$ . We added this introduction on Line 340-341 in Section 3.2.

P17 Fig. 8. Check the y-axis label of panel (a)

Response: We deleted Figure 8, according to the comments of reviewer 2.

P17 L361 It would make life easier for the reader if the column differences were expressed as a percentage.

Response: We deleted Figure 9 and the discussion about Figure 9, according to the comments of reviewer 2.

P18 Fig. 9. Include the parameters of the linear regression

Response: We deleted Figure 9, according to the comments of reviewer 2.

P18 L380 The meaning of the last sentence is unclear. Perhaps: "This is one of the few..."

Response: We modified the sentence to "This is one of the few reports about the detection of CFC-11 and CFC-12 columns and their tempo-spatial variations in China".

P19 L404 Start a new paragraph for the St. Petersburg comparison.

Response: We modified the part of conclusion.

P19 L407 It would be good to go on to describe the emission source differences

Response: We modified the part of conclusion.

References: There are some inconsistencies in formatting of the references, e.g. the use of capitalised journal and article titles, which should be rectified.

Response: We modified the format of all references.

**References**

Montzka, S. A., Dutton, G. S., Yu, P., Ray, E., Portmann, R. W., Daniel, J. S., Kuijpers, L., Hall, B. D., Mondeel, D., Siso, C., Nance, D., Rigby, M., Manning, A. J., Hu, L., Moore, F., Miller, B. R., and Elkins, J. W.: An unexpected and persistent increase in global emissions of ozone-depleting CFC-11, Nature, 557, 413-417. https://doi.org/10.1038/s41586-018-0106-2, 2018.

Steck, T.: Methods for determining regularization for atmospheric retrieval problems, Applied Optics, 41, 1788-1797. https://doi.org/10.1364/ao.41.001788, 2002.

Zhang, Y., Yang, W., Huang, Z., Liu, D., Simpson, I., Blake, D. R., George, C., and Wang, X.: Leakage Rates of Refrigerants CFC-12, HCFC-22, and HFC-134a from Operating Mobile Air Conditioning Systems in Guangzhou, China: Tests inside a Busy Urban Tunnel under Hot and Humid Weather Conditions, Environmental Science & Technology Letters, 4, 481-486. https://doi.org/10.1021/acs.estlett.7b00445, 2017.

Zhen, J., Yang, M., Zhou, J., Yang, F., Li, T., Li, H., Cao, F., Nie, X., Li, P., and Wang, Y.: Monitoring Chlorofluorocarbons in Potential Source Regions in Eastern China, Atmosphere, 11. https://doi.org/10.3390/atmos11121299, 2020.

---

## Author Comment (AC2)

**Reviewer #2**

**General comments:**

The authors used Tikhonov regularization to retrieve CFC11 and CFC 12 at Hefei station, China. They looked at the trend and seasonal cycle during 3 years for CFC 11 and 5 years for CFC 12. Although they mention new retrieval method in the abstract, they followed a very well know method of Tikhonov regularization. They also compared trends and the averaged retrieved profile shape with the ACE satellite as well as one NDACC station in St. Petersburg, Russia. Overall, there is a great value in creating independent ground data. However, the presentation and discussion of results needs to be improved. The information content of the retrieved data is only adequate for retrieving the total column for CFC 11 and maybe two column layers for CFC 12. But authors investigated the surface level value which is not meaningful information from the retrieval. They discussed that optimal estimation method (OEM) is not able to truly retrieve CFC data from FTIR sensors which is not correct. Authors wrote in a way that there is no way to constrain the results in OEM, however using more complex covariance matrix this is very possible. Also, I think the manuscript would benefit from adding a few more NDACC station rather one NDACC station to have a more meaningful comparison and discussion of the results. I would recommend a major correction is needed before publication of this manuscript. More detail comments are provided below.

Response: We appreciate your constructive comments. The comments and proposed corrections have been taken into account and helped improving the paper. Each comment has been addressed as follows. There is an extensive discussion among the authors regarding how to revise the content. Line numbers refer to the revised manuscript (version without tracked changes).

Specific comments:

- The general motivation of this work needs to be improved. Retrieval data has one DOF for CFC 11 and two DOF for CFC12 which means they can provide information about total column (assuming that the sensitivity is up to dominant portion of the CFC profile) and two partial columns for CFC12 (using average kernels they should identify the most meaningful layers that can be retrieved). The current motivation assumes that FTIR retrieval can retrieval a detailed profile from surface to stratosphere, which is not possible based on the sensitivity of these measurement. The motivation of the study needs to be rewritten to clarify how the retrieved information adds to the satellite and in situ measurement and the value of data based on true sensitivity of the data.

Response: We revised the objective of this work on Line 111-113 in Section 1 as follows:

The objective of this paper is to obtain the CFC-11 and CFC-12 total columns from the solar spectra based on ground-based FTIR spectroscopy, and compare with the ACE-FTS satellite data, WACCM data and the data from other NDACC-IRWG stations (St. Petersburg, Jungfraujoch, and Réunion).

Retrieving CFC profile is named as of the main objective of the manuscript. However, considering the low DOF it seems one total column (or two partial columns for CFC12) can be retrieved.

Response: We replaced the CFCs profile with CFCs total columns as the main objective on Line 111-113 in Section 1.

- It is not clear why only 3 years of data is used for CFC 11 while 5 years for CFC 12. Authors should clarify this inconsistency in the periods and elaborate on how that could affect their conclusions. Moreover, trend analysis with only 3 and even five years of data is not a robust conclusion. If authors include more NDACC stations and use longer period for available data, then they could discuss the overall trend of all included stations, and how a few years of Hefei compares with recent years of other stations.

Response: The spectral range for retrieval of CFC-11 is 830-860 $cm^{-1}$, while the spectral range for retrieval of CFC-12 is 1160.2–1161.4 $cm^{-1}$. We replaced $CaF_2$ incoming light window with KCl window for FTIR spectrometer in December 2016, which increased the covering spectral range from greater than 1000 $cm^{-1}$ to greater than 700 $cm^{-1}$. So we can retrieve CFC-11 since then. The explanation is included on Line 128-130 in Section 2.1. In addition, we added the data from two NDACC stations for comparison in Section 3.3, but the two NDACC stations do not provide long-period data.

Page 5, ln 146 to ln150. Authors mentioned that they did not use the optical estimation method because of the high fluctuation in their results. However, they can use more constrains in OEM by incorporating more a complex covariance matrix in the retrieval to achieve a more restricted result. It is fine to use the Tikhonov regularization, but the discussion of paper is in a way that OEM is limited which is not true in there are many ways to constrain your results to prevent high fluctuations. You can find some good examples of more complex covariance matrix to constrain the OEM results in these papers and many more online

1. Shams, Shima Bahramvash, Von P. Walden, James W. Hannigan, and David D. Turner. "Retrievals of Ozone in the Troposphere an Lower Stratosphere Using FTIR Observations over Greenland." IEEE Transactions on Geoscience and Remote Sensing(2022).

2. Turner, David D., and W. Greg Blumberg. "Improvements to the AERIoe thermodynamic profile retrieval algorithm." IEEE Journal of Selected Topics in Applied Earth Observations and Remote Sensing12, no. 5 (2018): 1339-1354.

Response: We read through the papers you provided, and think the discussion about that OEM is limited is not true, so we deleted this description in Section 2.3 of the revised manuscript.

- There are multiple NDACC stations, it is not clear why data is only compared with St Petersburg? It looks subjective, rather than rigorous research to find relevant and meaning full stations to compare. What is the impact of transportation and local sources. I would recommend authors use multiple station data for comparison to provide a more detail context for their comparisons. Specially that seasonal cycle has a lag time in their cycle. Having a clear discussion on different sources and sinks could cause these differences.

Response: Since CFC-11 and CFC-12 are not standard NDACC species, the data from most NDACC stations are not uploaded to the NDACC data archive. Following your suggestions, we contacted the researchers at the Jungfraujoch and Réunion stations to get the data from the two sites (for some reasons,

only the recent-year data are provided). We added the data from the two NDACC stations for comparison in Section 3.3, and have a discussion on which cause the differences.

- Page 6, ln 146 to ln156. Your methodology is very similar to Polyakov et al, 2019. Please discuss if there is any difference in the method that you used. Otherwise, I would suggest to just reference their paper as the same methodology is used and there is no need to repeating the same information. Especially that authors did not show any of these matrixes in the plots. As suggested in later comments, adding plots of DOF, average kernel, and Jacobian matrix is a nice way to characterize the sensitivity of the measurements. You can add formulation of those variables to clarify their meaning. Instead use the formulation to elaborate the errors that used in the text.

Response: We deleted the repeated discussion in Section 2.3 as suggested, and added formulas of DOFs, average kernels and covariance matrices to explain their meanings on Line 169-175 in Section 2.3.

- Page 12, ln 250, authors investigate surface level CFC 11 and CFC 12 in multiple plots. The measured FTIR data has 1 DOF for CFC11, thus there is not enough information to extract the near-surface value. Because a profile is incorporated as a priori, there is a profile output, however, there is no meaningful information at all levels of the profile. That's why DOF and averaging are useful information to indicate the sensitivity of the retrieval and the vertical resolution of the results. All the investigation of surface level needs to be excluded. Instead, if the DOF and average kernel show that the measurement has adequate sensitivity to one tropospheric column, the authors can analyze that data.

Response: We deleted the discussions on near-surface value as suggested, due to the low DOFs.

Writing/presentational comment

- Hefei is not part of the NDACC. (Could be a great addition though)

Response: In the abstract and Line 120-121 in Section 2.1, we added the description that Hefei is a candidate NDACC-IRWG station now.

- page 3, ln 71 to 75. It is mentioned that Yi et al, 2021 used in situ measurement. Are these measurements still active? What is the in situ temporal resolution? Again, the text implies that the retrieval can provide surface-level information, which is not correct. The text needs to be updated. Moreover, authors can include those surface measurements in their plots to compare with local measurements. On page 3, ln 78 to 88 for each satellite, please include the vertical and spatial resolution of the retrieved CFC.

Response: There is no report shows that the in-situ measurements of Yi et al. (2021) are still active. The measurements of Yi et al. (2021) collected air samples continuously for 4 - 7 days each month. We added this description on Line 79. Because we deleted the discussion on surface concentration in the revised manuscript, we didn't make this comparison in the revised manuscript. The vertical spatial resolution of

each satellite and airborne measurement were added on Line 83-97 in Section 1.

- Page 4, ln 108. Add solar before FTIR remote sensing site.

Response: We did it as suggested.

- Page 4. Ln 108 to 117: a map of the study area can be very helpful, to illustrate the location of Hefei and other NDACC stations will be added to the study.

Response: We added a map of the study area in Fig. 1. The map illustrates the location of Hefei, St. Petersburg, Jungfraujoch, St Denis and Réunion stations.

- Page 4, ln 123. It is not clear if the authors used the monthly climatology of CFC as the prior or if they used a specific prior for each month that they retrieved. (12 different profiles for each year of each gas.) also, it would be helpful to write the WACCM spatial resolution that is used in the study. I suggest the look at How the monthly variability and cycle of seasons of the received data is similar or different from prior information that is used.

Response: Here we used the averaged profiles calculated from the monthly profile of WACCM v6 at the Hefei station as the a priori profile. The WACCM v6 spatial resolution is $0.95\,° \times 1.25\,°$. We added the description on Line 139-140 and Line 341, respectively.

We ever made comparison between monthly variability and seasonal cycle of WACCM data and FTIR data, and the results are shown in the Figure as below. However, it can be seen that the monthly variability of FTIR and WACCM data is very different and there is no meaning information, so we didn't show this comparison in the manuscript.

[Figure]

[Figure]

- Page 8, ln 225. De Maziere et al., 2018 did not talk about the trends. Clarify which citation is related to each part of statement in this line.

Response: We corrected this citation error.

- Page 13, ln 260. The reasons behind the one-month phase delay need to be clarified. Look at other datasets (grand measurements in particular) and investigate if this delay is persistent there.

Response: Besides Hefei station, only St. Petersburg station provided both total column and profile of CFC-11 and CFC-12 data. Therefore, we compared the difference between monthly variation of total column and near-surface (about 200m) VMR for CFC-11 and CFC-12 at the St. Petersburg station, as shown in the Figure below. It can be seen that the monthly variations of total column and near-surface VMR at the St. Petersburg site are not the same, especially for CFC-12. For CFC-11, the lowest near-surface VMR occurring in April, is one-month later than March, when the lowest total column occurs, but both the maximum occur in August. For CFC-12, both the maximum total column and near-surface VMR occur in October, but the minimum near-surface VMR is July and that of total column is in March.

[Figure]

Te et al. (2016) found that there was a time lag of two months between surface VMR and total column of CO in Paris and Jungfraujoch, and described this time delay might be caused by different emission patterns. Therefore, the inconsistency between the monthly variation of near surface concentration and the total column for the target gas is persistent. However, we deleted the discussion on near surface VMR for our FTIR measurements as suggested, so the discussion on this phase delay was deleted in the revised manuscript.

- Page 18, ln 365. The seasonal cycle in St. Petersburg happens in fall which contradicts your explanation of the seasonal cycle in Hefei on page 13, ln 265. "In addition, more use of air conditioning and other refrigeration equipment in summer, and foams releasing more CFCs 265 at high temperatures lead to high concentrations of atmospheric CFCs." Authors need to further explain the seasonal cycle and its subjectivity to locations especially by adding more stations to the study it would be interesting to see how they vary and if that could lead to an interesting conclusion.

Response: We added the comparison of the seasonal cycle at four stations in Section 3.3. For the seasonal cycle of CFCs, emissions are one of the influencing factors, so we modified the description of reasons for seasonal variations at the Hefei site on Line 274-283 in Section 3.1.

Fig

- Add a figure of study location as well as selected NDACC data stations.

  Response: We added a map of locations of selected NDACC and Hefei stations in Fig. 1.

- Page 7. Fig 1. The total column average kernel is not very easy to comprehend. I would suggest including the DOF profile, mean averaging kernel profile, and the Jacobian matrix presentations to fully characterize the retrieval information.

  Response: We added the mean averaging kernel profile and the DOFs profile in Fig. 2(c) (d) and Fig. 3(c) (d). We don't know how to plot the Jacobian matrix, so we didn't add the Jacobian matrix.

- fig 3, and fig 5 (must check all the plots) axis has ccl2f2 on their axis while the caption says CFC. The same acronym should be used.

  Response: We made the acronym the same for all plots in the paper.

- Fig 5. Please include the averaged seasonal cycle from WACCM and ACE for the same period to show what information this study is bringing to the table.

  Response: We ever tried to study the seasonal cycle of ACE data. Unfortunately, due to the lack of observations in tropical and subtropical areas, ACE data within 27 °N–37 °N, 107 °E–127 °E are concentrated in February and May, most data are missing in January, March, June, July, November and December. Therefore, it is difficult to do the seasonal cycle analysis for ACE-FTS data. The seasonal cycle of WACCM data is not displayed, because of the large difference with that of FTIR.

Fig 6 and fig 4. the information content of these measurements is not sufficient to have meaningful surface value from the FTIR retrieval to investigate the results.

  Response: We deleted this part in the revised manuscript.

- Fig 8. It is not clear what information is depicted in this plot and what research questions are targeted here. There is so much subjectivity in two-point FTIR retrieval especially when they are located this far apart. I suggest removing this figure instead create some harmonic analysis of time series based on monthly CFC data for each station (as suggested before at lead 4 sites that are distributed in a different location) and discuss how the harmonic time series are similar or different.

  Response: We deleted Fig. 8 as suggested, and added comparisons of monthly CFC-11 and CFC-12 for four stations, shown in Fig. 7 and Fig. 8 of the revised manuscript.

Other modifications:

Response: In Section 3.1 of the revised manuscript, we modified the fitting formula slightly, and used

first-order polynomial and three harmonic terms to simulate CFC-11 and CFC-12 time seires, the equation $F(t)$ become:

$$F(t) = a + b \cdot t + \sum_{k=1}^{3}(c_{2k-1}\cos(2\pi kt) + c_{2k}\sin(2\pi kt))$$

where $a$ is the intercept, $b$ represents annual trend, and $c_1$ to $c_6$ represent sin/cosine harmonic term coefficient. The annual decreasing rate is obtained by the parameter $b$.

References

Te, Y., Jeseck, P., Franco, B., Mahieu, E., Jones, N., Paton-Walsh, C., Griffith, D. T., Buchholz, R. R., Hadji-Lazaro, J., Hurtmans, D., and Janssen, C.: Seasonal variability of surface and column carbon monoxide over the megacity Paris, high-altitude Jungfraujoch and Southern Hemispheric Wollongong stations, Atmospheric Chemistry and Physics, 16, 10911-10925. https://doi.org/10.5194/acp-16-10911-2016, 2016.

Yi, L., Wu, J., An, M., Xu, W., Fang, X., Yao, B., Li, Y., Gao, D., Zhao, X., and Hu, J.: The atmospheric concentrations and emissions of major halocarbons in China during 2009-2019, Environmental Pollution, 284, 117190. https://doi.org/10.1016/j.envpol.2021.117190, 2021.